# Are Torque-Induced Bone Texture Alterations Related to Early Marginal Jawbone Loss?

**DOI:** 10.3390/jcm11206158

**Published:** 2022-10-19

**Authors:** Tomasz Wach, Małgorzata Skorupska, Grzegorz Trybek

**Affiliations:** 1Department of Maxillofacial Surgery, Medical University of Lodz, 113 Zeromskiego Str., 90-549 Lodz, Poland; 2Department of Oral Surgery, Pomeranian Medical University in Szczecin, 70-111 Szczecin, Poland

**Keywords:** dental implant, torque, marginal bone loss, intraoral radiographs, radiomics, texture analysis, bone remodeling

## Abstract

The reason why marginal bone loss (MBL) occurs after dental implant insertion without loading has not yet been clearly investigated. There are publications that confirm or reject the notion that there are factors that induce marginal bone loss, but no research investigates what exactly occurs in the bone surrounding the implant neck. In this study, 2196 samples of dental implant neck bone radiographs were analyzed. The follow-up period was 3 months without functional loading of the implant. Marginal bone loss was evaluated in relation to the torque used during the final phase of implant insertion. Radiographic texture features were also analyzed and evaluated. The analyses were performed individually for the anterior and posterior part of the alveolar crest in both the mandible and maxilla. After 3 months, an MBL relation with higher torque (higher than 40 Ncm; *p* < 0.05) was observed, but only in the lower jaw. The texture features Sum Average (SumAverg), Entropy, Difference Entropy (DifEntr), Long-Run Emphasis (LngREmph), Short-Run Emphasis (ShrtREmph), and discrete wavelet decomposition transform features were changed over time. This study presents that MBL is related to the torque value during dental implant insertion and the location of the procedure. The increasing values of SumAverg and LngREmph correlated with MBL, which were 64.21 to 64.35 and 1.71 to 2.01, respectively. The decreasing values of Entr, DifEntr, and ShrtREmph also correlated with MBL, which were 2.58 to 2.47, 1.11 to 1.01, and 0.88 to 0.84, respectively. The analyzed texture features may become good indicators of MBL in digital dental surgery.

## 1. Introduction

Presently, the most common procedure in oral surgery after wisdom tooth extraction are dental implants [1]. With the increasing number of dental implant placements, more and more post-operative complications also occur. One of the major complications is marginal bone loss (MBL) next to the dental implant neck [2]. Marginal bone loss is a condition where the bone surrounding the implant neck atrophies. It is affected by different factors, e.g., smoking, diabetes mellitus, vitamin D and 25-hydroxycholecaliferol level, implant placement technique, region of the jaw, and also torque during the surgical procedure [3,4,5,6]. MBL may occur after few years, but also after the first 3 months of healing. The osteotomy techniques used for implant placement may also differ and may affect MBL [7]. MBL that occurs after a few years may be a condition related to more factors, e.g., prosthetic restoration and gingival vertical and horizontal width. MBL may be associated with a high torque value [8,9] if the correct procedure steps have not been followed and the appropriate primary stability of the implant has occurred [10].

The visual assessment of the surrounding implant neck bone on radiographs may not be sufficient or reliable. Another way to evaluate radiographic images is to check how the radiographic textures change over time—the shade level of the pixels can be analyzed. This analysescan be used, for example, to check how the texture of the bone substitute materials have changed after several months, translating this change into healing process progress [11,12,13,14].

The first aim of this study is to check whether torque and the location of the implant have an influence on MBL and to determine whether there is a change of radiographic texture in the bone surrounding the implant neck. The second aim is to check how the radiographic texture near the implant neck changes over time and to determine the prognostic factors of MBL in image texture.

## 2. Materials and Methods

In this study, 2196 samples of neck area implants were included and analyzed. A total of 504 males and 496 females aged between 15 and 86 years old were included in the study. All patients had undergone the same surgical procedure, namely, bone-level dental implant placement under local anesthesia (4% articaine with 1:100,000 adrenaline, 3 M ESPE AG, Seefeld, Germany). The patients were divided into two groups depending on whether MBL occurred after 3 months or not, and also into a mandible and maxilla group where the anterior and posterior parts were distinguished:MBL appearance (YES) if MLB is >0MBL appearance (NO) if MBL is =0

The inclusion criteria were two-dimensional radiographs taken immediately after surgery and 3 months later, the measurement of torque value immediately after dental implant placement, and laboratory tests to check patients’ vitamin levels, ions, and hormones: parathormone (PTH), where the norm is 10 to 60 pg/mL; thyrotropin (TSH), where the norm is 0.23–4.0 µU/mL; calcium in serum (Ca^2+^), where the norm is 9–11 mg/dL; glycated hemoglobin (HbA1c), where the norm is <5%; and vitamin 25(OH)D3 (D3), where the norm is 31–50 ng/mL. Spine densitometry, where the T-score can be examined, was also considered. The T-score shows the ratio between the bone mineral density (BMD) of the examined patient and the average BMD for young patients. A normal value for normal bone is >−1.0, osteopenia is indicated by values between −1.0 and −2.5, and scores < −2.5 indicate osteoporosis. The exclusion criteria included a lack of X-rays, defective X-ray images in the visual assessment, lack of a clear torque value, and lack of laboratory tests. In this study, only patients with proper values from the laboratory tests were included.

Surgery was done under local anesthesia, Septanest + A 1:100.00, by one surgeon according to the recommended protocols. The healing process was carried out under a closed mucoperiosteal flap, unloaded in two-stage implants. The thickness of the soft tissue did not affect the healing process or MBL in the first stage of healing. Table 1 presents the implants used in this study and their technical features. The data were confirmed at www.spotimplant.com/en/dental-implant-identification, accessed on 5 March 2022.

Two-dimensional X-ray images were taken immediately after surgery (00M) and 3 months later (03M). Radiographs were taken using the DIGORA OPTIME radiography system (TYPE DXR-50, SOREDEX, Helsinki, Finland). The radiographs were taken in the standardized way [15] with the following parameters: 7 mA, 70 mV, and 0.1 s (the focus apparatus was from Instrumentarium Dental, Tuusula, Finland). Positioners were used to take images repeatedly, with the X-ray beam at a 90° angle to the surface of the phosphor plate. The texture of the X-ray images was analyzed in the MaZda 4.6 software, developed by the University of Technology in Łódź, Poland [16], to check how the features changed over the 3 months of observation. A limitation of the study is that the laboratory tests were not checked after 3 months.

The analyses were performed in a few steps: first, all of the X-rays were edited (leveled) (Figure 1); next, the MBL near the implant neck area was measured (Figure 2), and then the X-ray image was loaded into MaZda in a bitmap file format. Next, the region of interest (ROI) was marked near the neck on the mesial and/or distal side of the implant (5–6 mm height) (Figure 3a). The ROI was marked on the RTG image immediately after inserting the dental implant and after 3 months of healing (Figure 3b). Any bone loss after 3 months was evaluated through radiographic analysis, as the vertical differences between the implant platforms and the first bone contact with the implant surface. The ROIs were normalized (μ ± 3σ) to share the same average (μ) and standard deviation (σ) of optical density within the ROI. The selected image texture features—sum of squares (SumOfSqrs), sum of average (SumAverg), entropy, different entropy (DifEntr), long-run emphasis moment (LngREmph), and short-run emphasis moment (ShrtREmph)—in the ROIs were calculated for the reference bone and for the bone near the implant neck. The Haar wavelet decomposition (LH, HL, LL, HH) was also performed and statistically analyzed after 3 months of observation. All features were gathered from four angles: 0°, 45°, 90°, and 135° from done pixel and the average value was later calculated.
$$SumAverg=\sum_{i=1}^{2N_{g}}ip_{x+y}(i)$$
$$SumOfSqrs=\overset{N_{g}}{\underset{N_{g}}{\sum }}\cdot \overset{N_{g}}{\underset{j=1}{\sum }}{\left(i-\mu_{x}\right)}^{2}\;p\left(i,j\right)$$
$$Entropy=-\overset{N_{g}}{\underset{i=1}{\sum }}\overset{N_{g}}{\underset{j=1}{\sum }}p\left(i,j\right)\log\left(p\left(i,j\right)\right)$$
$$DifEntr=-\overset{N_{g}}{\underset{i=1}{\sum }}p_{x-y}\left(i\right)\log(p_{x-y}\left(i\right))$$
where Σ is sum, *N* is the number of levels of optical density in the radiograph, *i* and *j* are the optical density of pixels five-image-point distant one from another, *p* is probability, and log is logarithm [11].
$$LngREmph=(\overset{N_{g}}{\underset{i=1}{\sum }}\overset{N_{r}}{\underset{j=1}{\sum }}j^{2}p\left(i,j\right))/C$$
$$ShrtREmph=(\overset{N_{g}}{\underset{i=1}{\sum }}\overset{N_{r}}{\underset{j=1}{\sum }}j^{-2}p\left(i,j\right))/C$$
where Σ is sum, *N* is the number of series of pixels with density level *i* and length *j*. *N_g_* is the number of levels for image density (8 bits, i.e., 256 gray levels), *N_r_* is the number of pixels in series, *p* is the probability, and *C* is the coefficient, as below:$$C=\overset{N_{r}}{\underset{j=1}{\sum }}\overset{N_{g}}{\underset{i=1}{\sum }}p\left(i,j\right)$$

### Statistical Analysis

The Kruskal–Wallis test (to compare time-dependent alternations in medians) was applied for statistical analysis. Next, a multiple comparison procedure was used to determine which means were significantly different from the others. The method discriminates among the variables Fisher’s least significant difference (LSD) procedure. The difference was considered significant if *p* < 0.05. Stargraphics Centurion XVI (Statgraphics-StatPoint Technologies, Inc., The Plains, VA, USA) was used for the statistical analyses.

## 3. Results

### 3.1. Marginal Bone Loss Statistical Evaluation

The statistical evaluation revealed that the amount of marginal bone loss was between 0 and 8.05 mm after 3 months of observation; the average MBL for the mandible was 0.29 ± 0.98 mm and for the maxilla was 0.23 ± 0.91 mm, which were both statistically significant at *p* < 0.01. The MBL means for the anterior and posterior parts of the jaw were 0.32 ± 1 mm and 0.22 ± 0.91 mm, respectively, where *p* was lower than 0.01, which means that that was statistically significant.

### 3.2. Torque Statistical Evaluation

It was also noticed that when MBL occurred, higher torque was observed in the mandible group (mean 46.77 Ncm ± 14) than in the maxilla group (mean 40.5 Ncm ± 11.9), with statistical significance (*p* < 0.05); the torque value during the implantation procedure was between 5 Ncm and 90 Ncm (Table 2).

### 3.3. Torque Value and Marginal Bone Loss as a Dependency

Marginal bone loss is related to the torque value during the implant placement (CC = 0.06, R^2^ = 0.3%, *p* < 0.05). If the torque increases, then MBL occurs more often (Figure 4). The study also showed that the average torque in the group with detected bone loss was higher than in the group without bone loss (42.01 and 40.04, respectively, where *p* was lower than 0.05, which means that was statistically significant) (Figure 5 and Figure 6).

### 3.4. Implant Placement Region in Jaw

After splitting the implant samples into two groups, the mandible and the maxilla group, the statistical evaluation revealed that marginal bone loss only correlated with torque in the mandible group (*p* < 0.01), even though most samples with MBL in the maxilla group occurred with a torque higher than 40 Ncm (Figure 7). There was no correlation between the groups where augmentation was performed before implant placement (*p* > 0.05).

Not all of the analyzed texture features were statistically significantly related to the marginal bone loss (*p* < 0.05). A *p*-value lower than 0.05 was observed for the following features after 3 months of observation:SumAverg,Entropy,DifEntr,LngREmph,ShrtREmph,Wavelets 4 and 5.

Texture features where the *p*-value was higher than 0.05, which means that this is not statistically significant, were:SumOfSqrs,Wavelet 6—was not detected.

As the reference, textural features for the trabecular bone were analyzed. The basal values for SumAverg, Entropy, DifEntr, LngREmph, and ShrtREmph are presented in Table 3. 

### 3.5. SumAverg Changes

The SumAverg for the implant neck area was 64.21 ± 2.9 at 00M. After 3 months, the SumAverg for the implant neck area with MBL was higher than after implantation, presenting 64.35 ± 3.54 and 64.16 ± 3.85, where MBL was not detected, and this was statistically significant (*p* < 0.05). There was no statistical difference between the maxilla and mandible groups (*p* > 0.05).

### 3.6. Entropy Changes

The Entropy for the implant neck area was 2.58 ± 0.19, and after implantation changed to 2.47 ± 0.21 where MBL was detected and 2.52 ± 0.2 where MBL was not detected, which was statistically significant. Statistically significant (*p* < 0.01) changes were also noticed in the maxilla and mandible groups with MBL (2.58 ± 0.12 changed to 2.52 ± 0.16, *p* > 0.05; and 2.58 ± 0.14 changed to 2.42 ± 0.16, *p* < 0.01, respectively).

### 3.7. DifEntr Changes

The DifEntr after implantation at 00M near the implant neck area was 1.11 ± 0.16 and changed to 1.01 ± 0.15 with MBL, and to 1.04 ± 0.16 where MBL was not present (*p* < 0.05). The DifEntr in the mandible group with MBL was 1.07 ± 0.15 and changed to 0.95 ± 0.14, which was statistically significant (*p* < 0.05).

### 3.8. LngREmph Changes

The LngREmph value at 00M was 1.71 ± 0.57 and changed to 2.01 ± 0.55 in the area with MBL, and 1.97 ± 0.75 where MBL did not appear (*p* < 0.05). The LngREmph value in the mandible group changed from 1.74 ± 0.64 to 2.11 ± 0.75 and there was no statistically significant difference (*p* > 0.05).

### 3.9. ShrtREmph Changes

The ShrtREmph also changed significantly: at 00M it was 0.88 ± 0.05, and after 3 months changed to 0.84 ± 0.05 for the area where MBL appeared and 0.85 ± 0.06 for the implant neck area without MBL. The ShrtREmph in the mandible group where MBL was correlated with MBL changed from 0.88 ± 0.6 to 0.84 ± 0.06, which was not statistically significant (*p* > 0.05).

### 3.10. Wavelet Decomposition Changes

The value for WavEnLH_s-4 after implantation for the maxilla group was 142.92 ± 111.54 and changed to 118.04 ± 87.24, which was statistically significant (*p* < 0.01). In the mandible group with MBL, the WavEnLH_s-4 changed from 134.35 ± 86.67 to 112.18 ± 116.14, which was statistically significant (*p* < 0.05).

The WavEnLH_s-5 in the mandible group changed from 345.23 ± 203.47 to 212.15 ± 185.22, which was statistically significant (*p* < 0.05). The WavEnHH_s-5 in the mandible group statistically significantly changed from 61.28 ± 65.77 to 35.75 ± 38.44, where the *p*-value was lower than 0.05 (*p* < 0.05).

Interestingly, the Wavelets 6 disappeared after 3 months of observation in the whole group with marginal bone loss (in the mandible and maxilla groups). The Wavelets 6 were also not detected in the reference trabecular bone. This texture index may be an indicator for cortical bone or changes in structure during the observation (Table 4, Table 5 and Table 6).

The research also showed that implant design also has an impact on marginal bone loss near the implant neck. This study compared several design properties: insertion implant level, neck microthreads, body shape, and body threads (thread shape). Apex shape, apex hole, apex groove, and connection type were not taken into account, as in the authors’ opinion, these implant features do not have an impact on marginal bone loss in the early period of healing without exposing the implant for oral cavity conditions.

The statistical evaluation showed that MBL in the case of bone-level implants was 0.26 ± 0.97 mm, and for subcrestal implants was 0.09 ± 0.51 mm, which was statistically significant (*p* < 0.05).

Greater MBL occurred near the neck of implants without microthreads (0.31 ± 0.92 mm) than near the implants where microthreads were present (0.25 ± 0.94 mm), and this was statistically significant at *p* < 0.05.

The marginal bone loss for implants without body threads was higher (0.99 ± 0.77 mm) than for the implants where the body threads were V-shaped (0.15 ± 0.64 mm). MBL was also shown in the case of square-, buttress-, and reverse buttress-shaped body threads at 0.28 ± 0.93 mm, 0.67 ± 1.75, and 0.25 ± 0.97 mm, respectively. Statistical significance at *p* < 0.05 was noticed between the reverse buttress and no threads; between square threads and no threads; and between V-shaped threads and no threads.

The research also showed that the MBL in the case of straight-body implants was 0.20 ± 0.49 mm and in the case of tapered implants was 0.25 ± 0.95 mm. There was no statistical significance, as *p* > 0.05 (Table 7).

Taking into account torque as a factor that can lead to MBL, some implant design features were examined depending on the insertion region and the torque used (higher or lower than 45 Ncm). Titanium alloy, the level of implant placement, the presence of microthreads on the implant neck, implant body shape, and also the design of the threads on the implant body were checked.

It was noticed that significantly higher MBL occurred in the anterior part of the mandible when the torque was lower than 45 Ncm, where dental implants made of titanium alloy Grade 4 (mean 1.06 ± 0.94 mm) were used. There was no correlation between MBL and titanium alloy in the maxilla samples. In the rest of the samples, titanium alloy was not correlated with MBL due to the localization of the implant or the insertion torque.

The research shows that the level of implant placement has an impact in the case of maxilla and mandible implants. In the anterior of the maxilla, greater MBL was noticed according to the tissue implant level (0.97 ± 0.88 mm) and the smallest MBL was found where subcrestal implants were used (mean 0.24 ± 1.14 mm), but only in cases when the torque was higher than 45 Ncm. In mandible-inserted implants, a subcrestal location had an impact on MBL in cases where the torque was lower than 45 Ncm (mean 1.06 ± 2.14 mm). The level of implant placement had no influence in the maxilla, either for the anterior part with a torque lower than 45 Ncm or the posterior part regardless of the torque used; and for the mandible, the anterior part with a torque higher than 45 Ncm and the posterior part regardless of the torque used.

Higher MBL was noticed in the anterior part of the maxilla when torque higher than 45 Ncm was used for implants without microthreads (mean 0.82 ± 1.76 mm). The presence or absence of neck microthreads in the case of the mandible, the posterior part of the maxilla, and the anterior part of the maxilla with torque lower than 45 Ncm was not correlated with MBL.

The shape of the implant body was not correlated with MBL in the case of maxilla and mandible dental implants regardless of the torque used during the insertions.

The research also showed that in maxilla implants with buttress threads inserted in the anterior part with a torque higher than 45 Ncm, and in the anterior part of the mandible where implants with V-shaped threads on the implant body were inserted with a torque lower than 45 Ncm, the MBL was greater and was statistically significant (mean 2.19 ± 3.34 mm and 1.06 ± 2.24 mm, respectively).

## 4. Discussion

The question is: Does the torque value of dental implants affect early marginal bone loss after 3 months of healing? There are publications about crestal bone stability claiming that there is no relation between implant torque insertion value and marginal bone loss [17]. In this study, 2196 samples of implant neck areas were analyzed and proved that there is a statistical relationship between a torque higher than 40 Ncm and marginal bone loss after 3 months of healing. Due to the lack of prosthetic loading, it can be declared that high torque during implant insertion is the main surgeon-related factor of MBL near the implant neck after 3 months of healing. Additionally, it was presented that dental implants inserted with a torque higher than 40 Ncm in the lower jaw were more susceptible to marginal bone loss, even though there are studies indicating that there is no correlation between marginal bone loss in the maxilla vs. the mandible [18,19].

The radiological or clinical evaluation of marginal bone loss is not always possible using only visual assessment. Sometimes, changes in the morphological part of the bone or bone substitute materials are not visible. Tomasz Wach et al. and Kozakiewicz et al. show that changes overcome during the healing process can be detected using texture features, already on the level of pixels [12,14,20]. Some of the texture features (inter-pixel relation in the optical density environment) can also be a good indicator of bone structure changes near the implant neck after 3 months of healing.

SumAverg and Entropy change over 3 months. It can be noticed that the decreasing value of SumAverg in cases where MBL was not detected can be a sign that the tissue around the dental implant neck is similar to intact trabecular bone tissue. The increasing value of Entropy also shows that in the group without MBL, the tissue around the dental implant neck after 3 months is more similar to trabecular bone. Referring to the other publications, it can also mean that the texture feature values of tissue near the implant neck after 3 months of healing in the group where MBL was detected approach cortical bone texture feature values [13,14].

The decreasing values of DifEntr and increasing values of LngREmph texture features are further proof that the tissue around the implant neck in the group with marginal bone loss is not similar to the trabecular one. For confirmation, Kozakiewicz et al. also proved that increasing LngREmph texture feature values and decreasing values of DifEntr are one of the signs of corticalization, which is correlated with bone loss [21]. In cases where MBL was detected and was correlated with corticalization, more longitudinal objects were observed (increasing LngREmph) as well as a decrease in chaotic patterns (decreased DifEntr).

Texture feature values that become similar to the reference of cortical bone can be an indicator of MBL. The process called corticalization near the implant neck surface is related to MBL [21]. On the other hand, an increasing value of the Entropy feature can be one of the signs that chaotic patterns have increased. Greater entropy means greater chaos, which can be equal to the decrease in bone structure.

It can be noticed that the values of the texture features in the group where marginal bone loss appeared were similar to the texture features of the reference scale for cortical bone. It is also related to higher torque when placing an implant. This means that higher torque during implant placement may lead to the densification of tissue around the implant neck. Condensed bone can be the first step to cortical bone formation, which is correlated with MBL.

Wavelet decomposition may be a healing process indicator. It can show that the trabecular part of the bone becomes part of the structure surrounding the dental implants. It also indicates the changes of long and small objects: longitudinal or circular shades [14,22]. It was noticed in Wach and Kozakiewicz’s study [14] that the higher the scale of wavelet decomposition, the larger the object appears in the texture. The subbands HH and LL indicate circular shades, while LH and HL indicate longitudinal ones. The decrease in wavelets in scale 4 and 5 in subband LH in the mandible group where MBL was detected may be related to the disappearance of bone structure. The disappearance of the decomposition of wavelets in scale 6 in our research was interesting. Our research can lead to the conclusion that the wavelets in scale 6 are an indicator of the healing process that occur later than 3 months post-operation. Wavelets in scale 6 should be observed in the next observation period.

The subcrestal implant level placement looks promising. This study shows that the lowest MBL appeared near this kind of implant. It is likely that the level of implant placement is not the only factor to have an impact on MBL. There are also studies that show that implant level placement does not have an influence on MBL near the implants [6]. MBL before the loading and exposure of the implant may be a result of different stresses of the implant neck.

Implant design features and structural features may also have an impact on marginal bone loss regardless of the torque used or the region of insertion [23,24], or may have preservation influence on the bone around the implants [25]. Taking into consideration that torque was not correlated with marginal bone loss in the case of implants in the maxilla, the research shows a correlation between MBL and implant design in the anterior part of the maxilla where the torque was higher than 45 Ncm. This proves that marginal bone loss in the maxilla is not correlated with a high insertion torque, or high torque along with some other implant feature can result in greater MBL. In the implant samples analyzed in the lower jaw, dental implant design had an insignificant impact on MBL in cases where the torque was lower than 45 Ncm. This is further proof that insertion torque (higher than 45 Ncm) has an impact on MBL near the implant neck after the first 3 months of healing.

One limitation of the study is that the laboratory tests were not checked after 3 months. Another limitation is that the radiograph texture analyses were not compared to the histopathological examination of bone near the implant neck. Clinical marginal bone loss in mm was not carried out because of trauma—the authors checked the marginal bone loss before the second stage of treatment—before exposing the implant. There was also a limited number of samples, as not all patients came back after 3 months of healing and not all of the pictures taken were qualified after visual assessment for analyses. The BMI of the patients was also not taken into account. The research needs further evaluation of other local factors that could have an impact on marginal bone resorption.

## 5. Conclusions

Marginal bone loss is related to higher torque value during the implant placement procedure (higher than 40 Ncm torque was closely associated with MBL), but only in implants located in the lower jaw. The texture feature values that change over the healing process are closely related to the occurrence of MBL. It can be concluded that there are some texture features that can be used as indicators of problems near the inserted implants. The texture feature values (SumAverg, Entropy, DifEntr, and LngREmph) can indicate the likelihood of MBL occurrence. Haar wavelet decompositions should be observed in the next period of observation. Selected implant design features near the implant neck may have a positive impact on marginal bone, lower early bone loss associated with the neck microthreads and body threads. The level of implant placement also has an effect on early MBL. Additionally, the statistics verified the correlation between the design of the implant, insertion region, and torque, showing that cases where the implants were inserted with higher torque were more vulnerable to MBL near the implant neck in the early period of healing. This study is also the beginning of research into the correlation between texture features near the implant neck on the day of surgery and future bone loss.

## Figures and Tables

**Figure 1 jcm-11-06158-f001:**
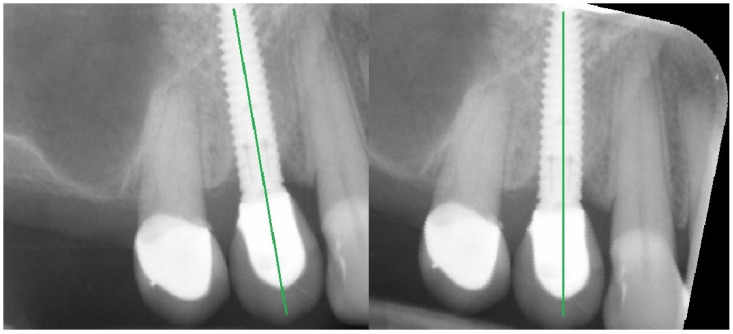
Geometrical alignment of radiograph image. The green line marked on the implants indicates the long axis of the implant.

**Figure 2 jcm-11-06158-f002:**
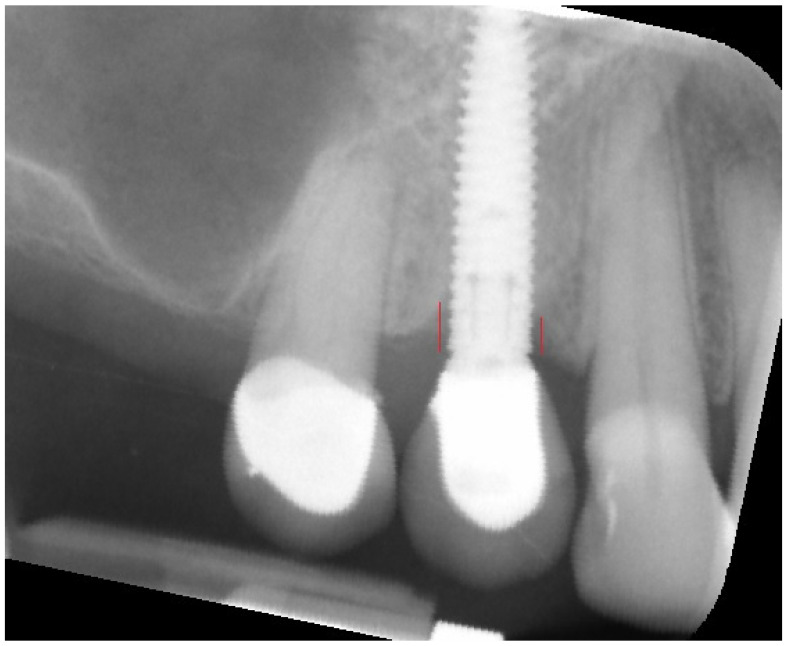
Measuring of marginal bone loss on the radiographic images. Red lines indicate the implant platform to the bottom of the bone loss cavity.

**Figure 3 jcm-11-06158-f003:**
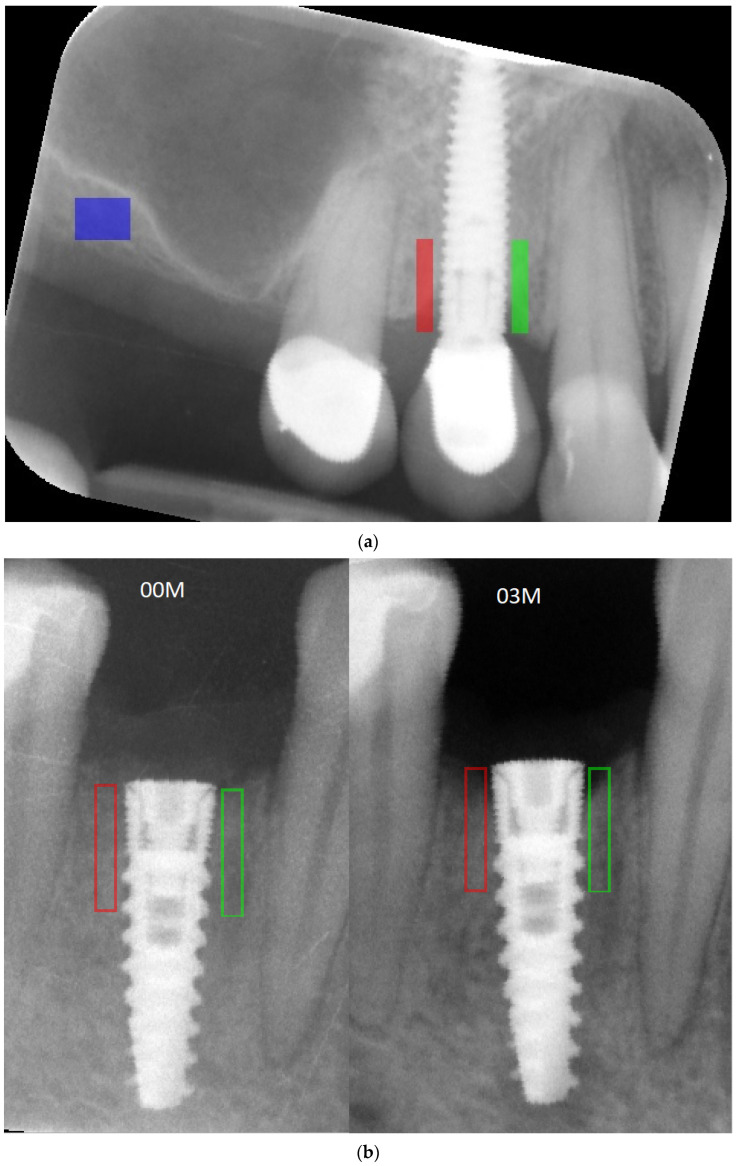
(**a**) Marking a ROI. ROIs were marked near the implant neck area. Green area—mesial implant neck area; red area—distal implant neck area; blue area—reference bone. Abbreviations: ROI—region of interest. (**b**) Marking a region of interest on RTG image immediately after inserting the implant and 3 months after the first stage of the healing process. Green area—mesial implant neck area; red area—distal implant neck area. At the bottom of the marked area on the right (03M), it can be noticed that MBL occurred and is analyzed. Abbreviations: MBL—marginal bone loss; 00M—0 months of observation; 03M—3 months of observation.

**Figure 4 jcm-11-06158-f004:**
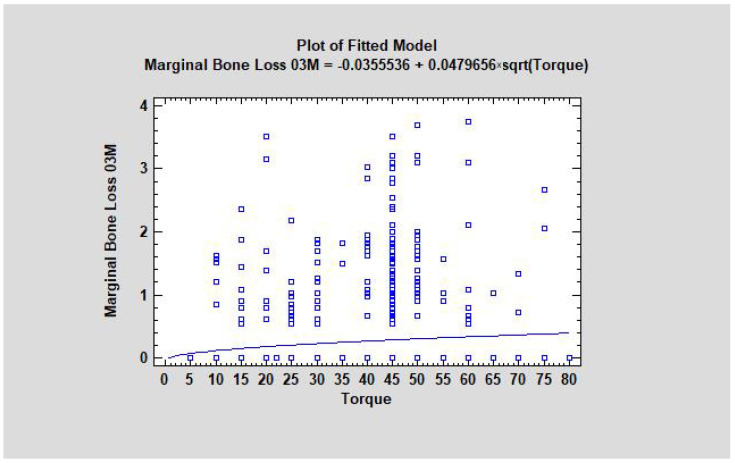
Dependence of marginal bone loss appearance from torque value after the dental implant insertion after 3 months of observation. The greatest number of marginal bone loss samples occurred in implants with a torque value equal to 45 Ncm. The higher the torque, the higher the MBL occurrence (with statistical significance). Abbreviations: MBL—marginal bone loss.

**Figure 5 jcm-11-06158-f005:**
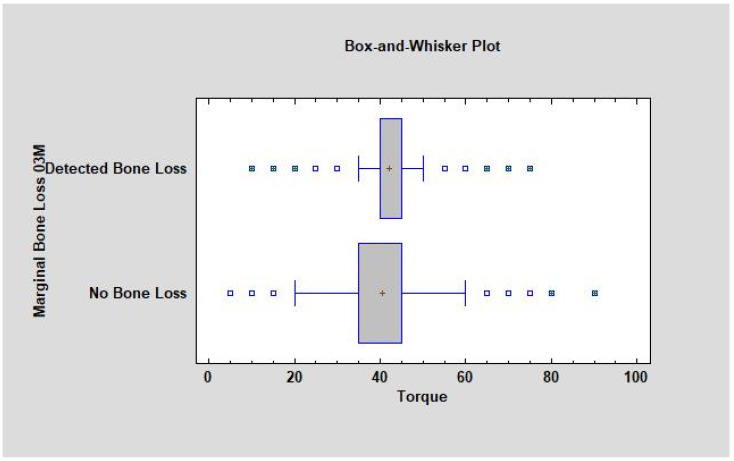
Dependence of average torque to marginal bone loss appearance after 3 months of observation. The mean where marginal bone loss was detected was higher than 42 Ncm. When the torque did not have an influence on MBL, the average torque was 40 Ncm or lower. Abbreviations: MBL—marginal bone loss.

**Figure 6 jcm-11-06158-f006:**
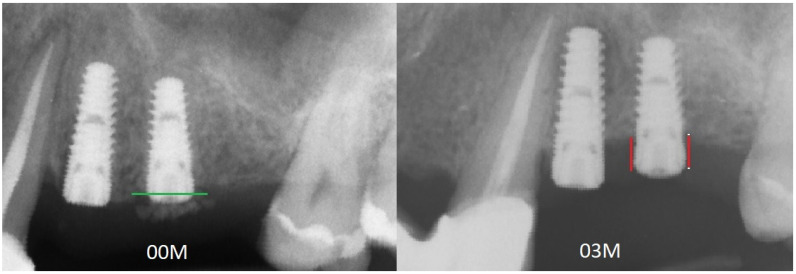
Figure presenting two radiographs. On the left is the radiograph image taken immediately after dental implant placement at 00M. The level of the bone at the beginning of the observation is marked by a green line. On the right is the RTG at 3 months of observation after implant placement. The red lines mark the marginal bone loss in relation to the previous bone level. Abbreviations: RTG—radiograph image; 00M—0 months of observation.

**Figure 7 jcm-11-06158-f007:**
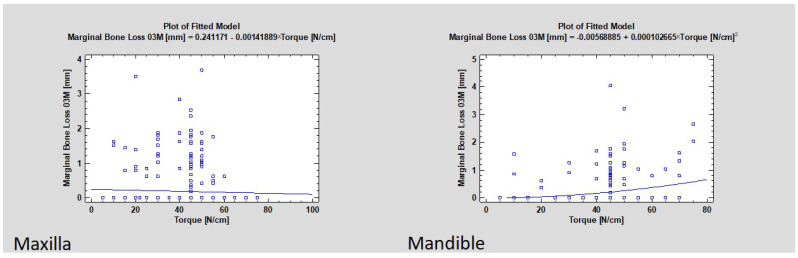
The graph on the left shows the dependence of marginal bone loss from dental implant insertion placement torque in the upper jaw (maxilla). There was no statistically significant difference (*p* > 0.05). The graph on the right presents the dependence of marginal bone loss on dental implant insertion placement torque in the lower jaw (mandible). There was statistical significance (*p* < 0.05).

**Table 1 jcm-11-06158-t001:** Names and features of implants used.

Implant Name	Titanium Alloy No.	Insertion Level	Connection Type	Connection Shape	Neck Shape	Neck Microthreads	Body Shape	Body Threads	Apex Shape	Apex Hole	Apex Groove
AB Dental Devices I5	Grade 5	Bone level	Internal	Hexagon	Straight	No	Tapered	Square	Flat	No hole	Yes
ADIN Dental Implants Touareg	Grade 5	Bone level	Internal	Hexagon	Straight	Yes	Tapered	Square	Flat	No hole	Yes
Alpha Bio ATI	Grade 5	Bone level	Internal	Hexagon	Straight	Yes	Straight	Square	Flat	No hole	Yes
Alpha Bio OCI	Grade 5	Bone level	Internal	Hexagon	Straight	No	Straight	No Threads	Dome	Round	No
Alpha Bio DFI	Grade 5	Bone level	Internal	Hexagon	Straight	Yes	Tapered	Square	Flat	No hole	Yes
Alpha Bio SFB	Grade 5	Bone level	Internal	Hexagon	Straight	No	Tapered	V-shaped	Flat	No hole	Yes
Alpha Bio SPI	Grade 5	Bone level	Internal	Hexagon	Straight	Yes	Tapered	Square	Flat	No hole	Yes
Argon Medical Prod. K3pro Rapid	Grade 4	Subcrestal	Internal	Conical	Straight	Yes	Tapered	V-shaped	Dome	No hole	Yes
Bego Semados RI	Grade 4	Bone level	Internal	Hexagon	Straight	Yes	Tapered	Reverse buttress	Cone	No hole	Yes
Dentium Super Line	Grade 5	Bone level	Internal	Conical	Straight	No	Tapered	Buttress	Dome	No hole	Yes
Friadent Ankylos C/X	Grade 4	Subcrestal	Internal	Conical	Straight	No	Tapered	V-shaped	Dome	No hole	Yes
Implant Direct InterActive	Grade 5	Bone level	Internal	Conical	Straight	Yes	Tapered	Reverse buttress	Dome	No hole	Yes
Implant Direct Legacy 3	Grade 5	Bone level	Internal	Hexagon	Straight	Yes	Tapered	Reverse buttress	Dome	No hole	Yes
MIS BioCom M4	Grade 5	Bone level	Internal	Hexagon	Straight	No	Straight	V-shaped	Flat	No hole	Yes
MIS C1	Grade 5	Bone level	Internal	Conical	Straight	Yes	Tapered	Reverse buttress	Dome	No hole	Yes
MIS Seven	Grade 5	Bone level	Internal	Hexagon	Straight	Yes	Tapered	Reverse buttress	Dome	No hole	Yes
Osstem Implant Company GS III	Grade 5	Bone level	Internal	Conical	Straight	Yes	Tapered	V-shaped	Dome	No hole	Yes
SGS Dental P7N	Grade 5	Bone level	Internal	Hexagon	Straight	Yes	Tapered	V-shaped	Flat	No hole	Yes
TBR Implanté	Grade 5	Bone level	Internal	Octagon	Straight	No	Straight	No threads	Flat	Round	Yes
Wolf Dental Conical Screw-Type	Grade 4	Bone level	Internal	Hexagon	Straight	No	Tapered	V-shaped	Cone	No Hole	Yes

**Table 2 jcm-11-06158-t002:** Average values for marginal bone loss and torque. Values are presented for mandible, maxilla, and for anterior and posterior areas of the jaw. Since the *p*-value is greater than or equal to 0.05, there is no statistically significant relation.

Feature	Marginal Bone Loss	*p*-Value for MBL	Torque	*p*-Value for Torque
Mandible	0.29 mm ± 0.98	*p* < 0.01	42.5 ± 12.67	*p* < 0.01
Maxilla	0.23 mm ± 0.91	*p* < 0.01	41.04 ± 12.7	*p* > 0.05
Anterior	0.32 mm ± 1	*p* < 0.01	43.15 ± 11.31	*p* < 0.01
Posterior	0.22 mm ± 0.91	*p* < 0.01	41.08 ± 13.07	*p* < 0.01

Abbreviations: *p*—the probability of obtaining test results at least as extreme as the results actually observed, under the assumption that the null hypothesis is correct.

**Table 3 jcm-11-06158-t003:** Reference texture feature values. Since the *p*-value is greater than or equal to 0.05, there is no statistically significant relation.

Texture Feature	Value	*p*-Value	Reference
SumAverg	63.22 ± 2.32	*p* < 0.05	Trabecular Bone
Entropy	2.70 ± 0.24	*p* < 0.05	Trabecular Bone
DifEntr	1.25 ± 0.12	*p* < 0.05	Trabecular Bone
LngREmph	1.53 ± 0.75	*p* < 0.05	Trabecular Bone
ShrtREmph	0.90 ± 0.05	*p* < 0.05	Trabecular Bone
WavEnLH_s-4	131.03 ± 94.39	*p* < 0.05	Trabecular Bone
WavEnLH_s-5	313.35 ± 213.69	*p* < 0.05	Trabecular Bone
WavEnHH_s-5	42.36 ± 44.35	*p* < 0.05	Trabecular Bone

Abbreviations: *p*—the probability of obtaining test results at least as extreme as the results actually observed, under the assumption that the null hypothesis is correct.

**Table 4 jcm-11-06158-t004:** Texture feature values at 00M, 03M, and for reference trabecular bone. Since the *p*-value is greater than or equal to 0.05, there is no statistically significant relation.

Texture Feature	Value at 00M	*p*-Value 00M	Value at 03M for the Area with MBL	Value at 03M for the Area without MBL	*p*-Value 03M	Reference Value for Trabecular Bone
SumAverg	64.21 ± 2.9	*p* > 0.05	64.35 ± 3.54	64.16 ± 3.85	*p* < 0.05	63.22 ± 2.32
Entropy	2.58 ± 0.19	*p* > 0.05	2.47 ± 0.21	2.52 ± 0.20	*p* < 0.01	2.70 ± 0.24
DifEntr	1.11 ± 0.16	*p* > 0.05	1.01 ± 0.15	1.04 ± 0.16	*p* < 0.01	1.25 ± 0.12
LngREmph	1.71 ± 0.57	*p* > 0.05	2.01 ± 0.55	1.97 ± 0.75	*p* < 0.01	1.53 ± 0.75
ShrtREmph	0.88 ± 0.05	*p* > 0.05	0.84 ± 0.05	0.85 ± 0.06	*p* < 0.01	0.90 ± 0.05

Abbreviations: 00M—0 months of observation; 03M—3 months of observation, MBL—marginal bone loss; *p*—the probability of obtaining test results at least as extreme as the results actually observed, under the assumption that the null hypothesis is correct.

**Table 5 jcm-11-06158-t005:** Texture feature values for mandible and maxilla groups after 3 months.

Texture Feature	Maxilla at 03M with MBL	Maxilla at 03M without MBL	*p*-Value for Maxilla	Mandible at 03M with	Mandible at 03M without	*p*-Value for Mandible
SumAverg	64.42 ± 0.91	64.37 ± 1.20	*p* > 0.05	64.76 ± 0.78	64.67 ± 0.89	*p* > 0.05
Entropy	2.52 ± 0.16	2.57 ± 0.14	*p* > 0.05	2.42 ± 0.16	2.50 ± 0.18	*p* < 0.01
DifEntr	1.05 ± 0.16	1.08 ± 0.15	*p* > 0.05	0.95 ± 0.14	1.01 ± 0.15	*p* < 0.05
LngREmph	1.89 ± 0.43	1.84 ± 0.48	*p* > 0.05	2.20 ± 0.68	2.11 ± 0.75	*p* > 0.05
ShrtREmph	0.85 ± 0.05	0.86 ± 0.05	*p* > 0.05	0.83 ± 0.6	0.84 ± 0.06	*p* > 0.05
WavEnLH_s-4	118.04 ± 87.24	122.93 ± 79.40	*p* < 0.01	112.18 ± 116.14	120.90 ± 74.04	*p* < 0.05
WavEnLH_s-5	304.59 ± 208.42	308.534 ± 268.32	*p* > 0.05	212.15 ± 185.22	287.54 ± 209.37	*p* < 0.05
WavEnHH_s-5	73.10 ± 65.97	63.89 ± 66.59	*p* > 0.05	35.75 ± 38.44	63.21 ± 73.00	*p* < 0.05

Abbreviations: 03M—3 months of observation; *p*—the probability of obtaining test results at least as extreme as the results actually observed, under the assumption that the null hypothesis is correct; MBL—marginal bone loss.

**Table 6 jcm-11-06158-t006:** Texture feature values for mandible and maxilla groups on the day of surgery.

Texture Feature	Maxilla at 00M before MBL Did Not Occur	Maxilla at 00M before MBL Occurred	Mandible at 00M before MBL Did Not Occur	Mandible at 00M before MBL Occurred
SumAverg	64.14 ± 0.15	64.16 ± 1.22	64.52 ± 0.94	64.71 ± 1.14
Entropy	2.61 ± 0.14	2.58 ± 0.12	2.56 ± 0.14	2.58 ± 0.14
DifEntr	1.15 ± 0.15	1.13 ± 0.12	1.07 ± 0.14	1.07 ± 0.15
LngREmph	1.68 ± 0.41	1.75 ± 0.52	1.77 ± 0.42	1.74 ± 0.64
ShrtREmph	0.88 ± 0.05	0.87 ± 0.05	0.87 ± 0.5	0.88 ± 0.6
WavEnLH_s-4	138.15 ± 94.13	142.92 ± 111.54	133.22 ± 86.73	134.35 ± 86.67
WavEnLH_s-5	331.45 ± 283.28	339.16 ± 300.11	324.12 ± 222.26	345.23 ± 203.47
WavEnHH_s-5	68.31 ± 88.00	69.94 ± 61.29	68.59 ± 75.42	61.28 ± 65.77

Abbreviations: 00M—the day of surgery; *p*—the probability of obtaining test results at least as extreme as the results actually observed, under the assumption that the null hypothesis is correct. MBL—marginal bone loss.

**Table 7 jcm-11-06158-t007:** Comparison of marginal bone loss depending on selected implant design features.

Compared Implant Design Feature	MBL	*p*-Value
Bone-level implant	0.26 ± 0.97 mm	*p* < 0.05
Subcrestal implant	0.09 ± 0.51 mm	
Neck microthreads	0.25 ± 0.94 mm	*p* < 0.05
Without neck microthreads	0.31 ± 0.92 mm	
Without body threads	0.99 ± 0.77 mm	*p* < 0.05
V-shaped threads	0.15 ± 0.64 mm	
Square threads	0.28 ± 0.93 mm	
Buttress threads	0.67 ± 1.75 mm	
Reverse buttress threads	0.25 ± 0.97 mm	

Abbreviations: MBL—marginal bone loss; *p*—the probability of obtaining test results at least as extreme as the results actually observed, under the assumption that the null hypothesis is correct.

## Data Availability

The data on which this study is based will be made available upon request at https://www.researchgate.net/profile/Tomasz-Wach, access date 16 October 2022.

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
