# Peer review of "Are Torque-Induced Bone Texture Alterations Related to Early Marginal Jawbone Loss?"

_jcm, 2022, doi:10.3390/jcm11206158_

Round 1

Reviewer 1 Report

Manuscript ID: jcm-1879705

Type: Article

Title: 
Are Torque-Induced Bone Texture Alterations Related with Early Marginal Jawbone Loss ?

Abstract: reformulate this section in line with the principles of abstract writing: aim, material methods, etc.

there are too many tables in the text, summarise some of them.

Introduction: Improve this section, explain what MBL is and talk about insertion torque and its function with references [PMID: 35770357 - PMID: 32098046 - PMID: 32475099]

Methods: explains how indoral radiographs were taken in a standard manner for each patient

Discussions: you must improve heavily by including the limits of the study.
Evidence the funding research part.

This manuscript must to be revalued after corrections

Author Response

Dear Reviewer,

thank You for Your detailed and insightful revision. Below are answers to Your comments. Corrects and additional text is improved into the manuscript and marked on green. We hope that now that it has been revised, our publication will meet your acceptance and requirements.

Yours Sincerely
Authors

Abstract: reformulate this section in line with the principles of abstract writing:

aim, material methods, etc. – Abstract was write according to Instruction for authors which are different “ The abstract should be a single paragraph and should follow the style of structured abstracts, but without headings: 1) Background: Place the question addressed in a broad context and highlight the purpose of the study; 2) Methods: Describe briefly the main methods or treatments applied. Include any relevant preregistration numbers, and species and strains of any animals used. 3) Results: Summarize the article's main findings; and 4) Conclusion: Indicate the main conclusions or interpretations” – if need I can reformulate abstract section.

there are too many tables in the text, summarise some of them. – Each table represents another part of the research. Each table ( even small one ) is specially prepared for the appropriate paragraph/part of the manuscript. If I will summarise some of them together, the tables in manuscript will be too big and not very easy to read. We hope that You will understand our position. We also hope that You will accept number of tables. Thank You.

Introduction: Improve this section, explain what MBL is and talk about insertion torque and its function with references [PMID: 35770357 - PMID: 32098046 - PMID: 32475099] – Thank You for the publications. Improved in the text.

Methods: explains how indoral radiographs were taken in a standard manner for each patient – the procedure was explained in the text. “Positioners were used to take images repeatably with 90° angle of x-ray beam to surface of the phosphor plate “

Discussions: you must improve heavily by including the limits of the study. – limited of the study improved at the end of discussion

Evidence the funding research part. – added as the additional attachement

This manuscript must to be revalued after corrections

Reviewer 2 Report

Dear authors, your study is interesting, however it does not include all relevant data for assessment of MBL, such as BMI, implant micro and macro-design, surface treatment and roughness (how many different implants were included in your study), mucosal thickness, surgical technique, etc.

You have even not listed that as the study limitations

If the authors can provide information relevant for the topic (implant systems-how many different systems were placed, implant geometry, surface treatment), preoperative bone density at the implant site, quantity (thickness) of cortical bone, etc., that will be really an interesting study. Study limitations must also be thoroughly discussed. Therefore my suggestion is that the authors re-write the manuscript and resubmit it.

Dear Editors and authors:

Here you can find my comments for the article:” Are Torque-Induced Bone Texture Alterations Related with Early Marginal Jawbone Loss”, sent for possible publication in the journal of clinical medicine

The article deals with marginal bone loss surrounding the implant neck after surgery before reopening the implants and healing abutment placement. Marginal bone loss surrounding the implant neck is analysed for correlation with the torque of implant insertion, jaw of insertion (mandible or maxilla) and anterior or posterior area of the jaw. MBL, Texture features SumAver, Entropy, DifEntr, LngREmph and discrete wavelet transform features were analysed on periapical radiographs, comparing post-insertion and 3-month radiographs.

Materials and Methods

Authors:The healing process was carried out under closed mucoperiosteal flap, unloaded in two-stage implants. Thickness of soft tissue did not affected on healing process and MBL in the first stage of healing.(64-65).

- This cannot be stated. The flap was closed after the surgery, but mucosal thickness can affect peri-implant MBL. It has even been advised in the textbooks to submerge implant 1-2 mm when mucosa is thin (thin biotype), i.e. when it’s height is less than 2 mm. The authors did not measure peri-implant mucosal thickness, and that is the limitation of the study.

Authors: Texture of X-ray images were analysed in MaZda 4.6 software, developed by University of Technology in Lodz (9), to check how the features changed over the 3 months of observation. A limitation of the study is that laboratory tests were not checked after 3 months. 75

– This can indeed influence the results. Did you choose subjects having laboratory values within normal boundaries. The statement must be stated in the discussion section, not here.

Authors:Selected image texture 84 features sum of squares ( SumOfSqrs ), sum of average ( SumAverg ), entropy, different 85 entropy ( DifEntr ), long-run emphasis moment ( LngREmph ), short-run emphasis mo- 86ment ( ShrtREmph ), in ROI’s were calculated for reference bone and for bone near the 87implant neck. The Haar wavelet decomposition (LH, HL, LL, HH) was also performed 88and statistically analyzed after 3 months of observation. All features were gathered in four 89angles 0°, 45°, 90° and 135° from done pixel, and average value was later calculated.

  • You have chosen reference bone away from the region of implant placement. eg. Fig 3a shows that you have chosen reference bone at the tuberosity region which is usually of much lower bone density and of different texture than the neighbouring bone in premolar region. Could you, please explain why?

  • Materials and Methods- other facts influencing periimplant bone loss which were not accounted:

Implant roughness and surface treatment,

Implant, especially implant neck micro and macro-design,

thread distance and length, angle of treads

mean roughness,

SL, SLA surface treatment, nanolayers

type, number, length, diameter?

  • How many different implants did you analyse? In your figures I can see at least 2 different implants with different threads and geometry at the implant neck level. That can influence MBL. At least you must list data about implants inserted and their differences in Mean Ra, neck thread design, thread geometry, helix angle, thread distance, etc., and must list that as the study limitation

  • Narrow-pitch" and "wide-pitch" implants influence osseointegration.

  • Secondly, you have no measure of periimplant bone left around implant after surgery – and that can influence MBL. Sufficient quantity of periimplant bone around implant neck is crucial.

  • Body mass index can also influence early MBL

  • In figure 6 00 it seems that some artificial bone was added or the implant was submerged – the data on how many implants were submerged is missing

  • Was the alveolar ridge levelled at the time of surgery. Leveling and flattening may remove cortical layer around implants which can also influence the rate of MBL

  • Different burs (different implant systems) usually drill 0.1 mm smaller diameter in the bone (bone bed) than the implant. However, in a low density bone implant bed can be thinner, so burs and implant bed dimensions also influence MBL

  • You must discuss about other possible factors influencing early MBL, or list the limitations of the study

  • Discussion

  • Does torque value of dental implants affect on early marginal 238
    bone loss after 3 months of healing – your study cannot answer this question. You must account more factors, and more importantly, you must list your limitations

-

Considering all of this, I recommend to reject paper in the current form, but if the authors can provide information relevant for the topic (implant systems-how many different systems were placed, implant geometry, surface treatment), preoperative bone density at the implant site, quantity (thickness) of cortical bone, etc., that will be really an interesting study. Study limitations must also be thoroughly discussed. Therefore my suggestion is that the authors re-write the manuscript and resubmit it.

Author Response

Dear Reviewer,

thank You for Your detailed and insightful revision. Below are answers to Your comments. Corrects and additional text is improved into the manuscript and marked on green. We are grateful for Your specialized revision and for Your interest of our work. We hope that now that it has been revised, our publication will meet your acceptance and requirements.

We counted and added some of the implant design properties You listed in revision. We would like also noticed that all implants where inserted with recommended protocols. We think that if implants are inserted by experienced surgeon and protocols are saved, than some of listed by You factors should not influence on MBL near implant neck after 3 months of healing. So the one of the crucial factor may be torque during the implant insertion. For example if the implant is wider than another and the neck area is properly prepared – diameter of the implant should not impact on early marginal bone loss. Also surface of the implant may have influence on osseointegration between implant and bone –we checked MBL near implant neck on implants which were osseointegrated. Study did not check the osseointegration of the implants – research checked marginal bone loss in case of implants which were osseointegrated with surrounding bone. If You say that the other factors of periimplant bone loss was not accounted we would like say that study did not check the periimplantitis. The healing was under mucosa. “Periimplantitis is a plaque-associated pathological condition that occurs in tussues around dental implants.” In our opinion peri-implantitis may occure after the exposure to oral factors.

Never than less, Your comment are valuable and  very helpful for our next research where implants were exposed to more factors.

Yours Sincerely
Authors

Materials and Methods

Authors:The healing process was carried out under closed mucoperiosteal flap, unloaded in two-stage implants. Thickness of soft tissue did not affected on healing process and MBL in the first stage of healing.(64-65). –  This cannot be stated. The flap was closed after the surgery, but mucosal thickness can affect peri-implant MBL. It has even been advised in the textbooks to submerge implant 1-2 mm when mucosa is thin (thin biotype), i.e. when it’s height is less than 2 mm. The authors did not measure peri-implant mucosal thickness, and that is the limitation of the study. - in our humble opinion thickness of soft tissue around the implant abutment and prosthetics has impact after the exposure/loading of the implant. In this study implants healing was closed for 3 months. Of  course that we can submerge implants – this is true – to avoid MBL affected by thin mucosa but this factor is important after the implant exposure. There is more solutions to get the appropriate mucosa thickness near the implant.

Authors: Texture of X-ray images were analysed in MaZda 4.6 software, developed by University of Technology in Lodz (9), to check how the features changed over the 3 months of observation. A limitation of the study is that laboratory tests were not checked after 3 months. 75 – This can indeed influence the results. Did you choose subjects having laboratory values within normal boundaries. The statement must be stated in the discussion section, not here. – correct – only patients with normal level in laboratory tests were included into this study. The limitations is that test were not checked after 3 months of observations and healing process. The statement is improved in the text.

Authors:Selected image texture 84 features sum of squares ( SumOfSqrs ), sum of average ( SumAverg ), entropy, different 85 entropy ( DifEntr ), long-run emphasis moment ( LngREmph ), short-run emphasis mo- 86ment ( ShrtREmph ), in ROI’s were calculated for reference bone and for bone near the 87implant neck. The Haar wavelet decomposition (LH, HL, LL, HH) was also performed 88and statistically analyzed after 3 months of observation. All features were gathered in four 89angles 0°, 45°, 90° and 135° from done pixel, and average value was later calculated.

  • You have chosen reference bone away from the region of implant placement. eg. Fig 3a shows that you have chosen reference bone at the tuberosity region which is usually of much lower bone density and of different texture than the neighbouring bone in premolar region. Could you, please explain why? – authors counted the refference value of texture feature to compare how the texture features are changing near the implant neck through the healing process . We can count texture feature for cortical bone, trabecular bone and for soft tissue away from the implant ( in our opinion the refference bone calculated near the implant could be disrupted by changes in bone tissue caused by the implant insertion neighbouring ). Bone density may be different taking into account the region of the jaw but texture features of trabecular bone is similar. Thanks to texture features we can detect different kind of tissues.  
  • Materials and Methods- other facts influencing periimplant bone loss which were not accounted: - This study did not check the periimplant bone loss. We were analysed the early marginal bone loss near the implant neck under the closed healing without exposure to the oral factors e.g. plague.

Reviewer 3 Report

1)     Abstract - authors stated that " Texture features SumAver, Entropy, DifEntr, LngREmph and discrete wavelet transform features were changed over time ". This statement is confusing. Please, use the full form of in the abstract.

2)     Introduction - Authors should increase the length of the introduction part as it will help the readers to understand the article.

3)     Result – The data presented in a good sequence but it could be improved with proper headings that will definitely help readers to understand it.

Author Response

Dear Reviewer,

thank You for Your detailed and insightful revision. Below are answers to Your comments. Corrects and additional text is improved into the manuscript and marked on green. We hope that now that it has been revised, our publication will meet your acceptance and requirements.

Yours Sincerely
Authors

1)     Abstract - authors stated that " Texture features SumAver, Entropy, DifEntr, LngREmph and discrete wavelet transform features were changed over time ". This statement is confusing. Please, use the full form of in the abstract. – Full form improved in abstract.

2)     Introduction - Authors should increase the length of the introduction part as it will help the readers to understand the article. – improved in the introduction

3)     Result – The data presented in a good sequence but it could be improved with proper headings that will definitely help readers to understand it. – improved in the results

Round 2

Reviewer 1 Report

Authors improved this manuscript quality.

Author Response

Dear Reviewer, thank You for Your suggestions that lead to better quality of our research.

Yours Sincerely
Authors

Reviewer 2 Report

Dear Editors, the authors tried to improve the manuscript, however there are some major issues. The statistical analysis was made separately between maxilla and the mandible, anterior and posterior parts, different dental implant neck and body designs, inserting implants at bone level or submerged, and also conical and hexagonal internal connection (for which I don't see the reason to compare before loading). The results may be by chance, as the variables may be correlated, which has not been done in the study. There are no proofs for adjustment of correlated variables, or analysis of all factors together which could contribute to the significance of  MBL differences. The professional statistician must be engaged. However, if the authors can improve the article thoroughly, I recommend the resubmission.

Author Response

Dear Reviewer, thank You again for all suggestions. Thanks to Your detailed review and all comments our research become better and more useful for clinicians. I hope that improvements we did will be correct and will finally meet with Your approval.

Yours Sincerely
Wach Tomasz